# Nanoscale resolution of microbial fiber degradation in action

**Meltem Tatli[1†], Sarah Moraïs[2†], Omar E Tovar-Herrera[2], Yannick J Bomble[3], Edward A Bayer[2,4], Ohad Medalia[1,2]\*, Itzhak Mizrahi[2]\***

[1]Department of Biochemistry, University of Zurich, Zurich, Switzerland; [2]Faculty of Natural Sciences, Ben-Gurion University of the Negev, Beer Sheva, Israel; [3]Biosciences Center, National Renewable Energy Laboratory, Golden, United States; [4]Department of Biomolecular Sciences, The Weizmann Institute of Science, Rehovot, Israel

**Abstract** The lives of microbes unfold at the micron scale, and their molecular machineries operate at the nanoscale. Their study at these resolutions is key toward achieving a better understanding of their ecology. We focus on cellulose degradation of the canonical *Clostridium thermocellum* system to comprehend how microbes build and use their cellulosomal machinery at these nanometer scales. Degradation of cellulose, the most abundant organic polymer on Earth, is instrumental to the global carbon cycle. We reveal that bacterial cells form 'cellulosome capsules' driven by catalytic product-dependent dynamics, which can increase the rate of hydrolysis. Biosynthesis of this energetically costly machinery and cell growth are decoupled at the single-cell level, hinting at a division-of-labor strategy through phenotypic heterogeneity. This novel observation highlights intrapopulation interactions as key to understanding rates of fiber degradation.

## Editor's evaluation

The premise behind this manuscript is timely and of interest to a broad scientific community working in the field of microbial recycling of cellulosic biomass. It provides a useful link between the occurrence and molecular aspects of the bacterial 'machinery' named cellulosome, and physiological traits of the same bacteria when grown on micro-crystalline cellulose. The key claims of the manuscript are well supported by the data, and the approaches used are thoughtful and rigorous.

**\*For correspondence:**
omedalia@bioc.uzh.ch (OM);
imizrahi@bgu.ac.il (IM)

[†]These authors contributed equally to this work

**Competing interest:** The authors declare that no competing interests exist.

## Introduction

Plants represent the largest reservoir of biomass on the planet (*Bar-On et al., 2018*). Hence, plant cell-wall degradation is essential for carbon recycling, environmental homeostasis, and of major significance to waste management and mammalian gut ecosystems. Bacteria are central players in this process, as some are capable of deconstructing cellulosic fibers into simple sugars that fuel microbial growth and fermentation cascades essential for nutrient cycling as well as their use in industrial processes (*Himmel and Bayer, 2009*). The intricate chemical composition of the plant cell wall is the major barrier to biomass recycling, and there is a long ongoing scientific and global interest to unravel how such microorganisms achieve the complex task of fiber degradation.

Several anaerobic bacteria have evolved energy-effective strategies to solve the challenge of fiber breakdown through the biosynthesis of multienzymatic complexes called cellulosomes (*Artzi et al., 2017*). Cellulosomes are massive multimodular protein machineries, composed of numerous enzymes attached via complementary cohesin–dockerin interactions to enzyme-integrating subunits called 'scaffoldins' (*Bayer et al., 2004*; *Fontes and Gilbert, 2010*; *Lamed et al., 1983b*). Genomes of the

different cellulosome-producing bacterial species can encode up to 200 different types of dockerin-bearing proteins (notably enzymes) which bind tenaciously to the cohesins of the scaffoldin subunit. The resultant multiprotein complexes, heterogeneous in their enzyme content, are highly efficient in deconstructing plant cell-wall polysaccharides. Although extensive research has been dedicated to investigate cellulosome activity and its composition in various cellulosome-producing bacteria (*Artzi et al., 2017*), how cellulosomes are structured and organized on the bacterial cell wall in situ remains largely unknown (*Bayer et al., 2009*). Understanding how these complexes interact with their substrates while they are situated on the bacterium would therefore contribute immensely to our comprehension and control of such an efficient enzymatic strategy. Moreover, the question of how and whether the molecular organization of these complexes on single cells is interconnected with functional and regulatory aspects of cellulosomes at the population level remains unanswered and is essential to our understanding of both the fiber-degradation process and these canonical enzymatic complexes.

Conventional methods have identified structural features of individual cellulosomal modules, as well as crucial interface residues that are responsible for the specific tenacious intermodular cohesin–dockerin binding between these definitive cellulosomal modules (*Adams et al., 2006*; *Carvalho et al., 2007*; *Carvalho et al., 2003*; *Nash et al., 2016*; *Noach et al., 2010*; *Noach et al., 2009*; *Salama-Alber et al., 2013*; *Wojciechowski et al., 2018*). In addition, larger cellulosome fragments have been structurally studied by complementary approaches, that is, crystallography, small-angle X-ray scattering, computational modeling, and negative-staining transmission electron microscopy (*Bomble et al., 2011*; *Bule et al., 2018*; *García-Alvarez et al., 2011*; *Hammel et al., 2005*; *Mayer et al., 1987*). Additional studies have revealed some of the structural features of a cellulosome fragment composed of five separate cellulosomal modules from three different proteins (*Adams et al., 2010*; *Currie et al., 2012*; *Smith et al., 2017*). More recently, truncated recombinant minicellulosome complexes, from the canonical and most-studied cellulosome-producing bacterium *Clostridium thermocellum*, comprising a trivalent truncated scaffoldin and three copies of the Cel8A cellulosomal endoglucanase, have been analyzed (*García-Alvarez et al., 2011*). These in vitro studies, focused mainly on specific enzymatic and structural components of the complex, without the presence of the microbial cells on which the cellulosomal complexes are situated. Pioneering studies published in the 1980s, using electron microscopic imaging, have provided a two-dimensional glimpse into the massive size and organization of the cellulolytic machinery on the bacterial cell surface of *C. thermocellum* (*Artzi et al., 2017*; *Bayer and Lamed, 1986*; *Lamed et al., 1983b*; *Madkour and Mayer, 2003*; *Mayer et al., 1987*). Nevertheless, these studies employed dehydration and chemical fixation procedures, which thus restrict the structural information and characterization of these bacterial cell-wall systems.

Hence, structural and ecological insights into the in situ fiber-degrading process at the molecular scale, while these complexes are in their active state, remains unknown. To this end, we exploited current technological advances that combined cryo-electron microscopy (cryo-EM) (*Bai et al., 2015*) and cryo-electron tomography (cryo-ET) (*Beck and Baumeister, 2016*; *Weber et al., 2019*) together with conventional approaches of microbial genetics, physiology, and biochemistry. Using these complementary approaches, we gained insights into the in vitro and in situ structure of the cellulosomal enzymes, from the single-enzyme level to the distribution of cellulosome complexes across single cells in a population, to reveal how cellulosome structure and distribution may be connected to gene expression and regulation at the population level.

## Results

### Cellulosomes surround the bacterium as an extensive layer of complexes at constant distance from the cell wall

To examine the detailed organization of the cellulosome complexes in situ, we applied cryo-EM and cryo-ET imaging to intact *C. thermocellum* bacterial cells. Low-magnification images of the bacteria indicated a typical view of the elongated cells, measuring a few micrometers in length and 500–600 nm in thickness (*Figure 1A*), which represents a suitable size for in toto analysis of bacteria by cryo-ET (*Ortega et al., 2020*). Closer inspection of these bacteria confirmed the presence of a dense layer surrounding the bacterium (*Figure 1A*, arrowhead). Slices through typical cryo-tomograms

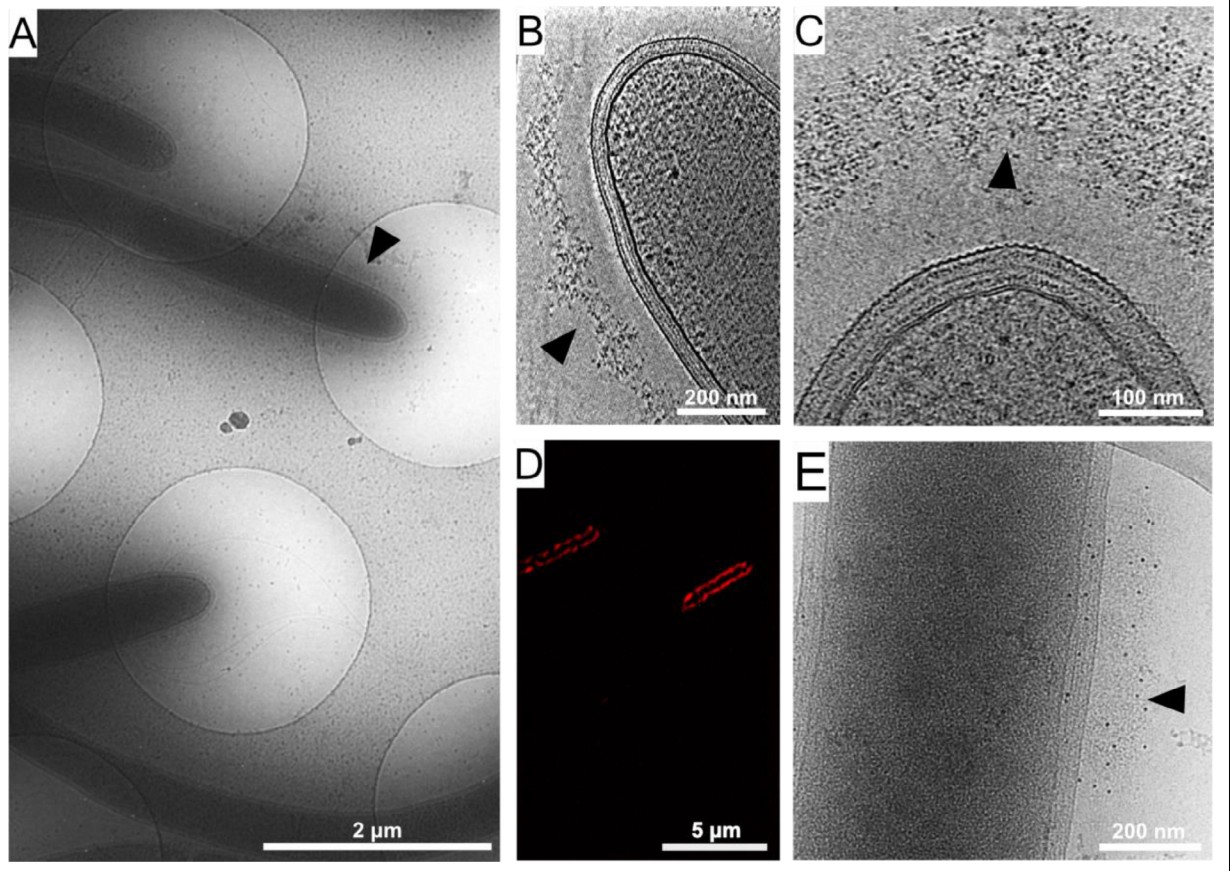

**Figure 1.** Cellulosome organization around *C. thermocellum* bacterial cells. (**A**) An image of *C. thermocellum* cells with surrounding cellulosomes on a cryo-EM grid at low magnification. The black arrowhead indicates the gray hue that forms a layer of extracellular cellulosome complexes surrounding the bacterium. At this magnification, the layer seems homogeneously thick. (**B, C**) Cellulosome complexes of two different bacteria at different magnifications. In (**B**), cellulosomes form a condensed layer which varies in thickness (arrowhead). At higher magnification in (**C**), the tomographic slice reveals the cellulosomes in greater detail (arrowhead). The density of the cellulosome layer reflects the protein occupancy within the layer. (**D**) Immunofluorescence confocal microscopy image of *C. thermocellum* bacteria reveals the position of the cellulose-binding module 3a (CBM3a) component of the CipA (ScaA) scaffoldin around the cells, using primary anti-CBM3a antibodies, detected with secondary goat anti-rabbit antibodies, conjugated with Alexa 594. (**E**) The extracellular density (arrowhead) detected by cryo-EM was verified to be cellulosomes. Immunogold labelling of cellulosomes, applied to localize the CipA scaffoldin using anti-CBM3a antibodies, indicates that the density detected around the bacteria corresponds to cellulosome complexes (***Video 1***, red arrowheads). The colloidal gold appears as black dots (arrowhead), while control experiments showed no gold nanoparticle signal around the bacteria (***Figure 1—figure supplement 2***).

The online version of this article includes the following figure supplement(s) for figure 1:

**Figure supplement 1.** Cellulosomes emanate from the cell wall toward the densely packed cellulosome layer.

**Figure supplement 2.** Cellulosomal protein layers around *C. thermocellum* cells.

(***Figure 1B,C***) revealed discrete globular protein densities in the extracellular layer and densities that emanate from the bacterial S-layer to this extracellular protein layer (***Figure 1—figure supplement 1***).

To confirm that the extracellular density layer detected by cryo-ET is composed of cellulosome assemblies, we utilized confocal immunofluorescence imaging (***Figure 1D***) and immunogold labeling with specific antibody targeting of the cellulose-binding module 3a (CBM3a) component of the CipA (ScaA) scaffoldin (***Figure 1E*** and ***Video 1***), the major defining constituent of the *C. thermocellum* cellulosome complex. The results revealed an extensive layer of cellulosome structures around these bacteria with varied densities at different levels (***Video 2***). Gold nanoparticles were detected at the densities around the cells, indicating the position of CBM3a within the cellulosome assemblies (***Figure 1E***, ***Figure 1—figure supplement 2A, B***, and ***Video 1***).

To further verify that the densities detected correspond to the cellulosomal fiber-degradation machinery, we analyzed two mutant strains of *C. thermocellum*, which either lacked the CipA scaffoldin

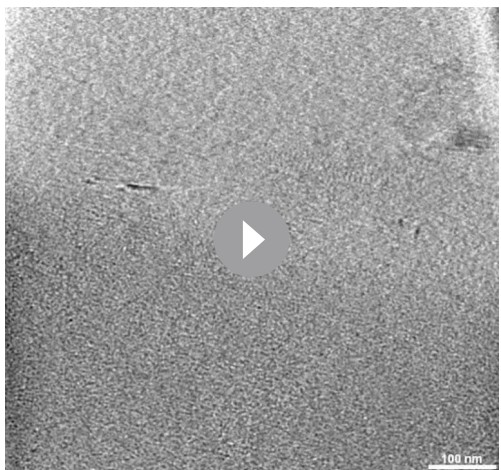

**Video 1.** The position of CBM3a revealed by immunogold labeling. The positions of CBM3a were identified by labeling with 6-nm gold clusters (red arrowheads). The protein was localized within the cellulosome layer.

https://elifesciences.org/articles/76523/figures#video1

subunit or lacked all of the anchored scaffoldins (knockout mutants DSN11 and CTN7, respectively) (*Xu et al., 2016*). Cryo-ET analysis of these mutant bacteria indicated complete absence of extracellular densities (*Figure 1—figure supplement 2C–F*) as compared to the wild-type cells. The above experiments suggest that the densities detected (*Figure 1*) correspond to the cellulose-degradation machinery of *C. thermocellum*.

Interestingly, cryo-tomograms of intact *C. thermocellum* cells revealed a characteristic gap which was not visible in chemically fixed samples between the S-layer and the cellulosome layer, with an average distance of 64 ± 17 nm (*Figure 2—figure supplement 3*), while the thickness of the cellulosome-rich layer was 65 ± 16 nm (*Figure 2—figure supplement 3*), measured using bacterial cultures that were cultured for 24 hr. Density emanating from the bacterium to the cellulosome layer was detected, which presumably indicated the anchorage of the cellulosome to the cell wall (*Figure 1—figure supplement 1*, arrows). The gap between the S-layer and cellulosome layer is reminiscent of the previously described 'contact corridors' (*Bayer and Lamed, 1986*), which were proposed to contain anchoring proteins, for example, OlpB (ScaB), that include exceptionally lengthy intermodular linking segments (*Bayer and Shimon, 1998*; *Madkour and Mayer, 2003*; *Shoham et al., 1999*).

## Cellulosome interactions with its substrate

How does the cellulosome interact with its cellulose substrate? Unprecedented insight into cellulosome–substrate interactions around the bacterial cell surface was obtained by utilizing volta-phase plate (VPP)-assisted cryo-ET analysis, which provides high-contrast imaging without the need for additional processing (*Danev and Baumeister, 2016*). We cultured *C. thermocellum* in the presence of microcrystalline cellulose (MCC) and were able to identify interactions between cellulose and the cell-associated cellulosomes. Strikingly, we observed the cellulosomes, which form a layer around the bacterium and envelop the cellulosic fibers (*Figure 2A,B* and *Video 3*). The cellulose was identified within the dense cellulosome layer, suggesting a flexible organization of the enzymes within the cellulosome layer while encompassing the cellulose substrate, presumably offering an advantage for its degradation (*Figure 2A,B* and *Video 3*). To further understand the enzyme–substrate interface and gain more insight into the density of enzymes around the substrate, we employed a template-matching procedure. For this procedure, we purified a recombinant form of the most expressed *C. thermocellum* dockerin-containing enzyme, which is crucial for enzymatic digestion of MCC, Cel48S (*Morag et al., 1991*; *Raman et al., 2011*; *Yoav et al., 2017*; *Zverlov et al., 2005*) and determined its structure at 3.4 Å resolution (*Figure 2—figure supplements 1 and 2*). The enzyme-matching structures were found in close proximity to the cellulose and appeared

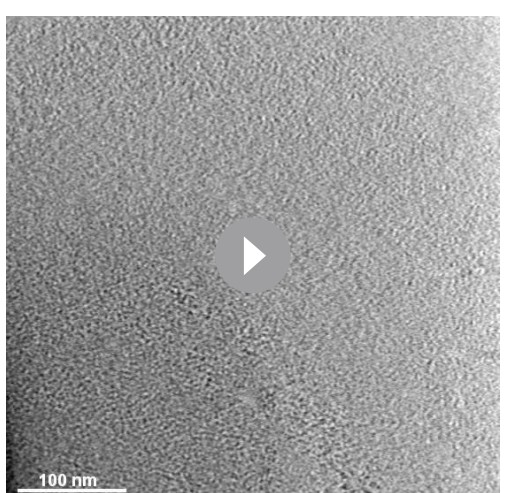

**Video 2.** A bacterium with high-density-cellulosome phenotype. Cryo-ET reveals the organization of the cellulosome structures.

https://elifesciences.org/articles/76523/figures#video2

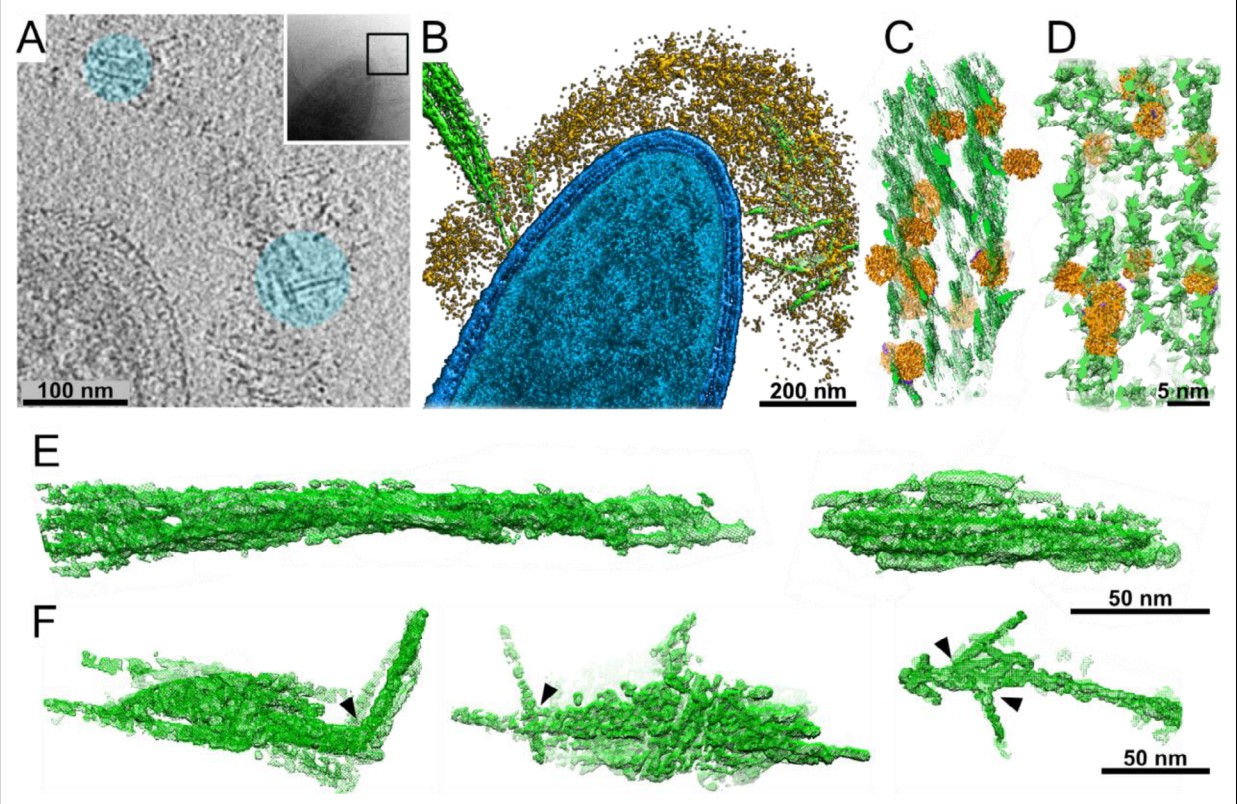

**Figure 2.** Interactions of cellulosomes with the cellulosic substrate. (**A**) An *x–y* projection through a tomogram of a bacterium grown on microcrystalline cellulose (MCC; cyan), ~1 nm in thickness, shows the MCC densities and globular protein densities. The area that was investigated is indicated in the inset, and individual projections of the region can be seen in the video (**Video 3**). (**B**) A surface-rendered view of *C. thermocellum* (blue) depicts the interactions of cellulosomes (yellow) with cellulose (green). Several cellulose fibers were deconstructed into their component microfibrils within the cellulosome layer. (**C**) A rendered model indicating the position of cellulosomal enzymes around a cellulose fiber, enzymes (orange) and cellulose (green) in situ. (**D**) A rendered model of purified Cel48S, in complex with a divalent truncated scaffoldin, bound to a cellulose fiber, enzymes (orange) and cellulose (green). Positions of enzymes (orange) in (**C, D**) were determined by template matching and resemble each other (**Figure 2—figure supplement 4**). (**E**) Surface rendering of purified MCC indicates a compact filamentous assembly. (**F**) The parallel packing of cellulose fibrils is altered (arrowheads) upon the interaction with cellulosomes. These surface-rendering views of cellulose fibers were isolated in silico from in situ tomograms where interactions with cellulosomes were observed.

The online version of this article includes the following source data and figure supplement(s) for figure 2:

**Figure supplement 1.** Structural determination of the Cel48S enzyme.

**Figure supplement 2.** Structure of Cel48S depicts the linker region.

**Figure supplement 2—source data 1.** Data collection and processing statistics of the cryo-EM structure.

**Figure supplement 3.** Position and thickness of the cellulosome layer.

**Figure supplement 4.** Representative cryo-ET projection slices of microcrystalline cellulose (MCC)–cellulosome interactions.

to localize between the cellulose chains, presumably, to increase the efficiency of degradation (**Figure 2C**). In order to better understand and validate our observation for the interactions between cellulosome and cellulose, we purified Cel48S together in a designer cellulosomal complex with a recombinant divalent truncated CipA scaffoldin that contains two cohesin modules (cohesin numbers 2 and 3) and the inherent CBM3a (see Materials and methods). This in vitro minicellulosome complex was incubated with MCC and analyzed by cryo-ET. The architecture found in this in vitro experiment (**Figure 2D** and **Figure 2—figure supplement 4C, D**) shows an overwhelming resemblance to that found in situ around the bacterium, further corroborating our findings (**Figure 2C**). Indeed, our analysis revealed that the structure of cellulose undergoes major organizational changes during the interaction with the cellulosomes, transforming from a well-ordered crystalline cellulose structure (**Figure 2E**) to deconstructed crystalline packaging (**Figure 2F**, arrowheads). It should be noted that at the resolution of individual tomograms, we cannot distinguish the Cel48S from the other large globular enzymes of

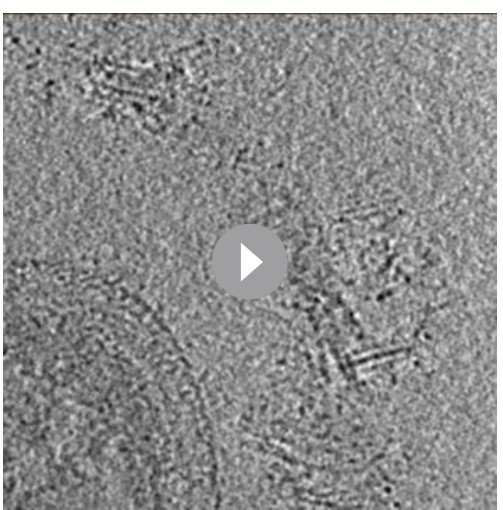

**Video 3.** The cellulosome layer accommodates crystalline cellulose. Globular enzymatic densities (orange) of the cellulosome layer were observed attached to the crystalline cellulose (green).
https://elifesciences.org/articles/76523/figures#video3

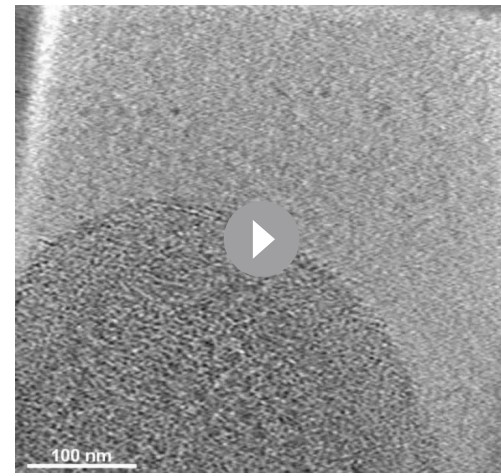

**Video 4.** A bacterium with low-density-cellulosome phenotype.
https://elifesciences.org/articles/76523/figures#video4

the cellulosome. Therefore, it is plausible that the depicted densities in our rendered view can be attributed the Cel48S enzyme as well as to additional cellulosome containing enzymes. Since the contrast of the globular cellulolytic enzymes is relatively high in comparison to the flexible rod-like appearance of scaffoldins, we could not reliably follow the densities of scaffoldins.

These observations provide snapshots of cellulose-degradation intermediates produced by the functional cellulosomal enzymes, located in the dense cellulosome layer around the bacterium. The findings further corroborated that our approach promoted the capture of real-time changes in cellulose organization and structure when interacting with cellulosomal complexes (*Figure 2A*, blue circles, *Video 3*).

## Dynamic isogenic phenotypic heterogeneity of cellulosome organization

The detachment of the cellulosomal machinery from the cell surface during culture growth has been reported since its initial discovery (*Bayer et al., 1985*; *Lamed et al., 1983a*; *Lamed et al., 1983b*), thus raising questions concerning the extent of this phenomenon and its single-cell distribution across the microbial population, as well as its ecological relevance. Indeed, while studying the structure of the cellulosomal machinery during cellulose degradation, we observed two distinct phenotypes of cellulosome organization on the bacterial cells, that is, high-density (raw data in *Figure 3—figure supplement 1A*, *Video 2*), and low-density organization (raw data in *Figure 3—figure supplement 1B*, *Video 4*). To further pursue this phenomenon, we quantified the single-cell distribution of the cellulosome on the bacterial cells by employing cryo-EM as a tool for structural analysis of individual bacterial cells within the *C. thermocellum* bacterial population. Our analysis revealed clear phenotypic heterogeneity within the same environment, under the same conditions, with distinct cellulosome density distributions representing two different cell phenotypes. The distribution of these two phenotypes in the stationary-phase culture shows a ratio of 1:5 (high versus low density).

Phenotypic heterogeneity is considered to be an important ecological strategy for microorganisms to cope with environmental fluctuations and could be mediated by gene expression, where the proportion of cells that express the related phenotype changes as a function of environmental cues (*López-Maury et al., 2008*). To explore this notion, we subjected a stationary-phase culture, containing a large proportion of low-density phenotype, to a fresh medium, containing cellulose as the sole carbon source, and quantified the ratio between the populations of the two phenotypes during cell growth (*Figure 3A*), using high electron-dose-exposure cryo-EM images. We analyzed 1160 individual cells, sampled from cultures at five time points after inoculation, which correspond to different population

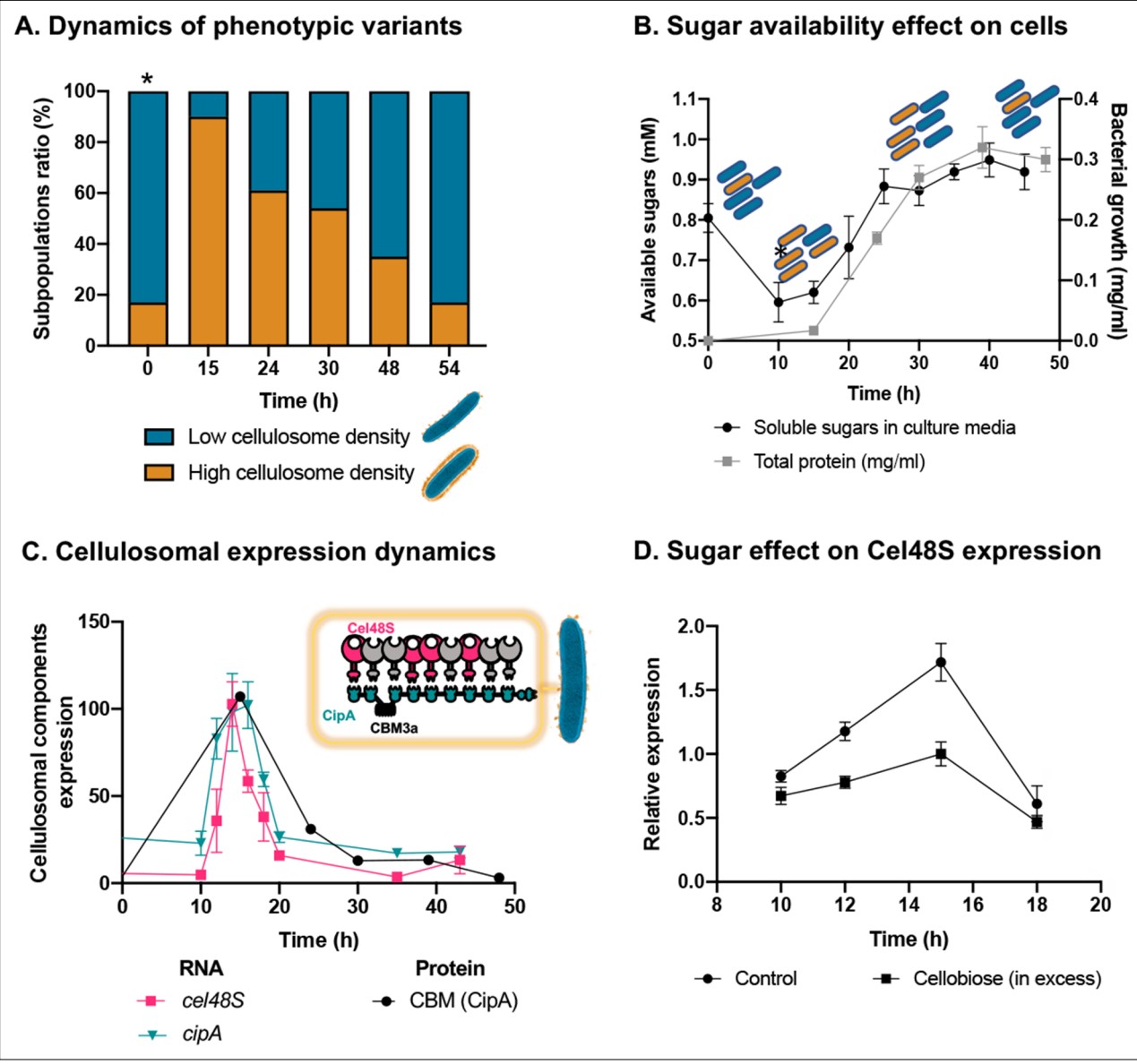

**Figure 3.** Dynamic isogenic phenotypic heterogeneity of cellulosome organization. (**A**) Dynamics of phenotypic variants in *C. thermocellum* subpopulations. A proportion of 5:1 (low density:high density) is observed during lag and stationary phases, while the high-density subpopulation is transiently dominant at 15 hr, gradually decreasing in time to the initial proportion. Time 0 hr was not measured, as it corresponds to the proportion of cells from the inoculum of time 54 hr. (**B**) Available reducing sugars detected in *C. thermocellum* cultures grown on cellulose over time (black line, in mg/ml for total protein). A decrease in soluble sugars (10–15 hr) is followed by a rise in their concentration, which is maintained stable until 50 hr. Blue and orange rod-like shapes represent the low- and high-density subpopulations, respectively, and provide a schematic view of the relative changes in their distribution across the different growth phases and the levels of available sugars. (**C**) Expression dynamics of cellulosomal components at the level of transcript (RNA) and protein expression (units represent normalized copies/cells divided by 20,000 or band intensity, respectively). Per-cell RNA transcript copy number of *cel48S* (pink line) and *cipA* genes (blue line) from a culture of *C. thermocellum* growing on microcrystalline cellulose. At the protein level, CBM expression per cell was evaluated by using anti-CBM3a antibodies detecting the CBM present in the CipA scaffoldin. Cellulosomal gene expression is observed as a single peak after 15 hr of growth of *C. thermocellum* in a medium containing microcrystalline cellulose. The schematic representation of the *C. thermocellum* cellulosome, highlighting the major components tracked in this study, is given as an inset. (**D**) Cellulosomal gene expression is controlled by sugar availability. Addition of high concentration of cellobiose (10 g/l) into a 10-hr culture of *C. thermocellum* negatively affects the expression of the main cellulosomal enzyme Cel48S (squares), compared to control (circles) as reported previously (*Morag et al., 1991*). In (**B–D**), error bars represent the standard deviation between three biological replicates.

The online version of this article includes the following figure supplement(s) for figure 3:

**Figure supplement 1.** High- and low-density cellulosomes.

**Figure supplement 2.** Overall enzymatic activity of the cellulosomal machinery during population growth.

growth phases (with an average of *n* = 232 cells per time point). Our analysis revealed that, upon exposure of the bacterial cells to the new environment containing only cellulose, a sharp increase in the ratio of the high-density phenotype occurred, after which a gradual decrease in its concentration was observed back to a 1:5 ratio (*Figure 3A*). Since soluble sugars, which are the product of the plant fiber degradation, are lacking at the beginning of culture growth, where only cellulose polymers are present as a carbon source, the findings raised the hypothesis that the soluble sugars, accumulated during population growth, would affect the ratio between the two subpopulations.

To address this hypothesis, we measured the availability of soluble sugars across the growth curve in the medium that contains cellulose as the sole carbon source. Our results revealed that the availability of soluble sugars derived from cellulose degradation is tightly connected to the ratio between the two subpopulations and culture growth stage (*Figure 3B*). At early stages of population growth, where the availability of soluble sugars is low, an increase in the high-density subpopulation occurs, whereas upon increase in sugar availability with population growth, the ratio of the low-density subpopulation increases (*Figure 3B*). Interestingly, in the late stages of growth, when the high-density subpopulation comprises only 20% of the overall population, the amount of the soluble sugar concentration remained stable and did not decrease even when the population growth reached its peak. This observation supports the notion that soluble sugar is provided in saturating amounts at this stage, and the 1:5 ratio of the high-density subpopulation is therefore maintained (*Figure 3*, 30–50 hr). These findings appear to reflect dynamic phenotypic variability, which infers that the accumulation of soluble sugars would be related to this phenomenon.

## Dynamic phenotypic variability is controlled by availability of soluble sugar

As our results suggest that the expression of the cellulosomal machinery changes during population growth and exposure to new environmental conditions, we proceeded to better understand cellulosomal expression dynamics and the factors that influence its regulation, within the context of our findings. To this end, we examined the expression dynamics and activity of the cellulosomal machinery during population growth after exposure to a fresh cellulose-containing medium.

Our results showed similar dynamics to that of the single-cell phenotypic analysis observed by cryo-EM: that is, a sharp increase in expression of the cellulosomal machinery upon exposure to the new environment, which was substantially reduced at the early stages of growth and then maintained at basal levels. This increase in cellulosome expression was corroborated at the RNA levels of the two major cellulosomal components, Cel48S and CipA (*Figure 3C*), thus corroborating previous reports (*Dror et al., 2003a*; *Dror et al., 2003b*; *Ortiz de Ora et al., 2018*; *Ortiz de Ora et al., 2017*), and at the protein level where we quantified the expression levels of CipA, using a specific antibody against its CBM3a component (*Figure 3C*). We also measured the total enzymatic activity, normalized per cell of the culture, which comprises both the enzymatic machinery on the cell surface and enzymes secreted into the extracellular environment. The total cellulolytic activity presents the same pattern, which comprises a sharp increase in activity during early growth, followed by a sharp decrease in later growth stages (*Figure 3—figure supplement 2*).

We next examined our hypothesis that soluble sugar availability resulting from cellulose degradation modulates the expression of cellulosomal components. To this end, we devised an experiment in which we added cellobiose, the major degradation product of cellulose, at the onset of cellulosomal expression (10 hr after exposure of the cells to cellulose). Addition of cellobiose resulted in a negative effect on Cel48S expression, the main cellulosome component (*Figure 3D*), thereby suggesting that the availability of soluble sugars as products from the cellulose-degradation process is one of the cardinal ways for controlling the expression of cellulosomal components (*Figure 4*).

## Discussion

In this work, we studied the plant fiber-degradation process at the scale in which it occurs, that is, micro- to-nanoscale level, from the individual microbes to the enzyme level, respectively. We selected *C. thermocellum*, one of the most efficient and well-characterized fiber-degrading microbes in nature. This bacterium deploys a multienzyme cellulosomal strategy that allows it to extract energy from highly resistant plant-fiber polymers, notably crystalline cellulose.

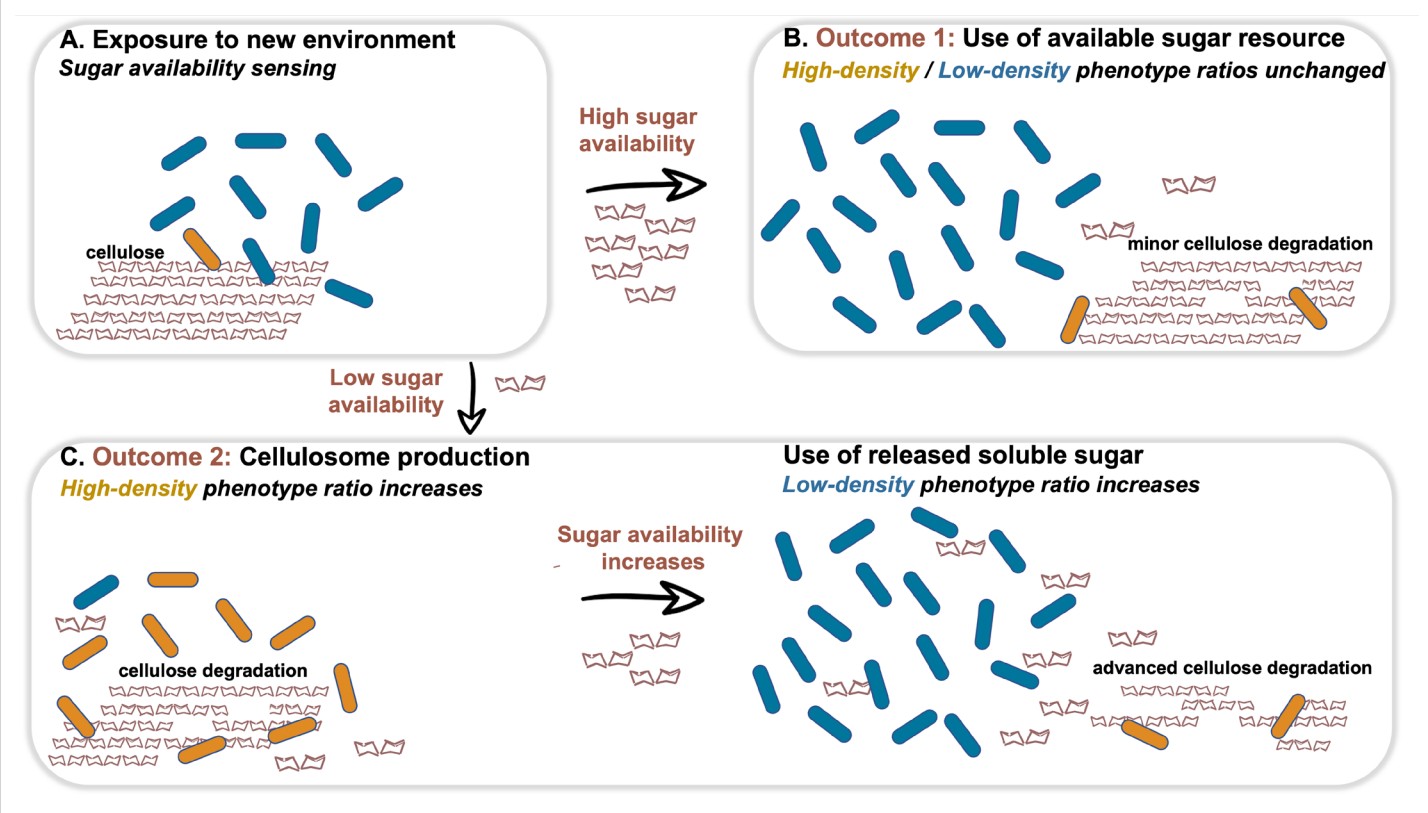

**Figure 4.** Dynamic phenotypic heterogeneity during cellulose degradation in the *C. thermocellum* population. The figure represents a schematic model of the findings revealed in our study. Blue and orange rod-like shapes represent the low- and high-density subpopulations, respectively. When *C. thermocellum* cells are exposed to a new environment containing cellulose (**A**), there are two possible outcomes depending on sugar availability. High sugar availability results in a basal level of cellulosome expression with no change in phenotypic heterogeneity of the population (**B**), where the high-density phenotype serves as a cellulosomal reservoir for future sugar-limiting conditions. In contrast, low sugar availability triggers the expression of the cellulosomal machinery (**C**), resulting in a major change in phenotypic heterogeneity of the population, where the high-density phenotype ratio increases and provides the present and future populations with high levels of sugar production from the cellulose substrate. This in turn increases the available sugar concentrations, arresting cellulosome expression and production resulting in an increase in the ratio of the low-density phenotype without the production of the high-cost enzymatic machinery, whereby the high-density phenotype returns to low ratios in the population (bet-hedging strategy). The secreted cell-free cellulosomes and their interaction with the cellulose substrate are not represented in the figure.

Cellulosomes are extraordinarily efficient cellulosic fiber-degradation machineries that have evolved to provide sugar metabolites for subsequent assimilation by the parent bacterial cell. Our exploration spanned from unique mechanistic insights into the way cellulosomes are distributed on the bacterial cell surface, the way their enzymes interact with and degrade plant-fiber polymers, and the way single cells employ phenotypic heterogeneity at the population level.

Structural analysis by cryo-ET of *C. thermocellum* cells provided a tool to study the structure of cellulosomes in a near-physiological state, to understand their distribution on single cells, and to inspect the interactions of the bacterium with cellulose fibers (*Figure 2A,B*). In addition, our structure-based approach served to generate physiological and ecological hypotheses.

Our study revealed that cellulosome complexes are arranged as a highly dense interconnected layer of protein around the bacterial cell at a distance of ~64 nm from the cell wall (*Figure 2—figure supplement 3*). Intriguingly, we could directly observe the cellulosomal machineries during their interaction with the plant-fiber polymers as they envelop them with their numerous enzymes and modulate the crystalline organization of the cellulosic substrate (*Figure 2E,F*). We also determined the structure of the dominant cellulosomal enzyme, namely the Cel48S cellobiohydrolase. Our data confirmed a previously published X-ray structure of this enzyme (*Liu et al., 2018*) and added new information regarding the amino acids that link the enzyme to its dockerin and on the interaction of this enzyme with cellulose. In addition, our data suggested a flexible organization of the enzymes within the

cellulosome layer while encompassing the cellulose substrate, presumably offering an advantage for its degradation. Such conformational adaptability of the cellulosome on the cellulose surface was also recently observed by atomic force microscopy (*Eibinger et al., 2020*).

Our structural approach led to the detection and visualization of two major subpopulations in the *C. thermocellum* culture under the same experimental conditions: one with cellulosomes covering the bacterial cells in very high density and another in which cellulosomes are nearly absent (*Figure 3— figure supplement 1*). We found that the ratio between these two phenotypes is dynamic and connected to the population growth stage and sugar availability (*Figure 3A,B*). After exposure to new environmental conditions lacking soluble sugars, the expression level of the cellulosomal machinery increased in the majority of the cells in the population, after which it was repressed in most cells of the population. Furthermore, our findings indicate that the expression of the cellulosome machinery is triggered upon exposure to low levels of soluble degradation products (e.g., cellobiose) and then inhibited by increasing sugar availability (*Figure 3B–D*). These findings add an additional dimension to the regulation of expression of the cellulosome machinery, which has been reported previously to be mediated by sensing of extracellular polysaccharides via alternative sigma factors (*Nataf et al., 2010*). In addition, it is interesting to note that the basal expression level of the cellulosome machinery at time 54 hr is 20% of its highest expression levels at time 15 hr (*Figure 3C*). This corresponds to the proportion of the high-density subpopulation in the overall population (20%) at times 0 and 54 hr (*Figure 3A*), thus suggesting that the basal RNA levels stem from the expression of the cellulosome machinery in the high-density cell subpopulations.

The distribution of cellulosome density on the cell walls exhibited an all-or-none phenotype, where cells were either highly decorated with cellulosomes or almost devoid of cell-surface cellulosomes, without observable distribution of intermediate phenotypes (*Figure 3—figure supplement 1A, B*), thus suggesting that cellulosome shedding occurs in an abrupt manner, similar to protein secretion, and not a gradual process during growth. Taking our overall findings into account, it is tempting to speculate that a division-of-labor strategy takes place in the *C. thermocellum* population. In this scenario, the high-density subpopulation expresses the cellulosome machinery comprising enormous amounts of proteins and consequent metabolic costs, thus limiting the resources that this subpopulation can invest in growth, while at the same time the low-density subpopulation does not pay these costs and can invest them in propagation. Hence, the low-density subpopulation balances the essential but costly high-energy process of expressing the cellulosomes by the high-density subpopulation.

Finally, our findings are consistent with a bet-hedging strategy in the *C. thermocellum* population, mediated by phenotypic heterogeneity, whereby the presence of diverse phenotypes adapted to different environments would spread the risk of varying environmental conditions (*Villa Martín et al., 2019*). Such a strategy would be particularly effective for organisms under fluctuating environmental conditions, where a proportion of individuals in the population pay an energetic/fitness cost by 'preparing' for changing environmental conditions. Similarly, in our case, although the soluble sugar levels are not limiting at stationary phase (*Figure 3B*), the high-density cell subpopulation, albeit at lower ratios, still expresses the costly cellulosomal machinery and would therefore provide the first line of defense for a future nutrient-poor and hostile environment containing only cellulose polymers.

The approach, whereby structural biology under near-physiological conditions was employed to understand microbial ecology, provides a pioneering view, at the micro- and nanoscales, into the unique organization of the cellulosome machinery on the bacterial cell surface, the way cellulosomes interact with their cellulosic substrate and how they are distributed and regulated across the bacterial population during the growth process.

## Materials and methods
### Expression of the Cel48S enzyme and scaffoldin protein
The plasmid for Cel48S expression was previously described (*Vazana et al., 2010*). The plasmid for expression of the truncated scaffoldin CipA (*Yaron et al., 1995*) was prepared by cloning the divalent Cohesin2–CBM3a–Cohesin3 region from the *C. thermocellum* genomic DNA using primers 5′-t tactCCATGGgctcCgacggtgtggtagta-3′ and 5′-ttgatCTCGAGtgtatctccaacatttactc-3′ and Phusion High-Fidelity polymerase (NEB, USA). The PCR product was restricted with NcoI and XhoI enzymes (FastDigest, ThermoScientific) and inserted into pET28a restricted with the same enzymes. Competent *E. coli*

BL21 Star (DE3) cells were used to express both proteins in 2 L cultures, using isopropyl-1-thio-β-D-galactoside(Fermentas UAB Vilnius, Lithuania) to induce expression. Cells were harvested by centrifugation at 3000 × g for 5 min. The Cel48S protein was purified using a Ni-NTA column (Qiagen, Hilden, Germany) as reported previously (*Caspi et al., 2009*) and further purified by size exclusion chromatography (Superdex 200 Increase, GE Healthcare Biosciences, Piscataway, NJ) in Tris Buffer Saline (TBS, 137 mM NaCl, 2.7 mM KCl, 25 mM Tris at pH 7.4) + 3 mM CaCl$_2$ + 2 mM β-mercaptoethanol buffer. The expected peak was eluted at 12 ml, and the molecular weight of the protein in the selected fraction was confirmed by sodium dodecyl sulfate–polyacrylamide gel electrophoresis (SDS–PAGE) and mass spectrometry. The truncated scaffoldin was purified using its affinity to phosphoric acid-swollen cellulose as described earlier (*Vazana et al., 2010*).

For assembling the Cel48S in complex with the divalent truncated scaffoldin, the Cel48S enzyme was added at a molar ratio 2:1 to the truncated scaffoldin protein and incubated for 2 h at room temperature with 10 mM CaCl$_2$.

## Sample preparation for cryo-EM and tomography

The concentration of Cel48S was diluted to ~0.35 mg/ml in ice-cold elution buffer, before being vitrified on carbon-coated Au grids (R0.6/1, 200 mesh; Quantifoil, Jena, Germany).

*C. thermocellum* DSM1313 cells were grown on GS2 medium (*Yoav et al., 2017*) containing either 8% cellobiose or 0.2% MCC as carbon source. The cells were grown anaerobically at 60°C. The cells grown on cellobiose were harvested at 24 hr, and the cells grown on crystalline cellulose were harvested at different time points, ranging from 15 to 54 hr of incubation (*Figure 3A*). Cells were centrifuged for 5 min at 5000 × g, washed briefly in TBS buffer, and resuspended in TBS prior to flash-freezing. A volume of 3 µl of cell sample was mixed with 3 µl of 10-nm fiducial bovine serum albumin BSA gold marker (Aurion, Wageningen, The Netherlands) and applied on glow-discharged copper EM grids, coated with a carbon mesh (R2/1, 200 mesh; Quantifoil, Jena, Germany). The grids were manually blotted for ~4–5 s from the reverse side and plunge frozen in liquid ethane by using an in-house plunger.

To analyze the organization of MCC (Sigma-Aldrich), dilute suspensions (0.04–0.0016 g/l in TBS) were prepared, 3 µl of the suspensions were mixed with 3 µl of 10 nm fiducial BSA gold marker (Aurion, Wageningen, The Netherlands) and applied on glow-discharged copper EM grids, coated with a carbon mesh (R2/1, 200 mesh; Quantifoil, Jena, Germany).

Cel48S interactions with MCC in vitro were studied as follows: A stock suspension of Avicel – MCC (Sigma-Aldrich) – was prepared at a concentration of 10 % (wt/vol) in a typical assay mixture of buffer (100 mM acetate buffer pH 5.0, 24 mM CaCl$_2$, 4 mM ethylenediaminetetraacetic acid). The MCC suspension and the protein solutions were mixed at a ratio of 1:8 (MCC:Cel48S-scaffoldin). Next, they were incubated at 37°C for 1 hr with continuous agitation, and the suspension was centrifuged at 3000 × g for 5 min. The supernatant fluids were deposited onto EM grids and plunge frozen as described above.

## Cryo-EM data acquisition

A total of ~2000 micrographs were acquired using Titan Krios (Thermo Fisher Scientific, Waltham, MA) equipped with Gatan energy filter and K2 summit (Gatan Inc, Pleasanton, CA). Using an underfocus range of 0.5–1.5 µm and the super resolution mode resulted in a pixel size of 0.425 Å. Exposures of 0.2 s for 12 s resulted with 60 frames and a total dose of ~67 e⁻/Å$^2$.

## Cryo-EM image processing

The micrographs were drift-corrected and binned once using Relion 3.0, resulting in a pixel size of 0.86 Å. A total of ~1100 manually selected particles were used to create a 2D reference for further automatic particle picking using Relion 3.0. The structural determination steps are described in *Figure 2—figure supplement 1* and *Figure 2—figure supplement 2—source data 1*. Initially, a few rounds of reference-free 2D classifications using ~363,000 particles resulted in an initial reference using Relion 3.0 by Stochastic Gradient Descent. 3D classifications and 3D refinements were performed by using the initial reference with the refined 3D classes. One the 3D-refined classes reached 3.6 Å resolution and was further improved by Bayesian polishing and CTF refinements. The final 3D structure was determined to be 3.38 Å resolution, using ~55,000 particles.

The obtained final model was compared to the pdb of the Cel48S exoglucanase (5yj6, *Liu et al., 2018*) by the fit-in-map option in Chimera, resulting in a correlation score of 0.87. The RMSD between 1 and 663 amino acids of two models were in 5.7 Å. The stoichiometry of the model was improved by real-space refinement in Phenix, and the final atomic structure was validated by MolProbity.

## Immunogold labeling of cellulosomes

The protocol was adapted from *Turgay et al., 2017*. A 1 ml sample of bacterial culture was washed in TBS as described above, and 5 µl of the washed cells were introduced to Au grids (R2/1, 200 mesh; Quantifoil). The grids were fixed in 4% paraformaldehyde (in TBS) for 5 min at room temperature and were quenched in 0.05 M glycine/PBS (purchased phosphate buffer saline) for 15 min at room temperature to inactivate aldehyde groups present after aldehyde fixation. Next, samples were blocked in blocking solution (5% BSA/0.1% cold water fish skin gelatin/PBS; Aurion, Wageningen, Netherlands) for 30 min at room temperature, and subsequently washed with the incubation solution (0.2% BSA-c [acetylated bovine serum albumin for eliminating the background]/PBS; Aurion). A volume of 20 µl of anti-CBM3a antibody (*Morag et al., 1995*) (1:1000 dilution in incubation solution) was incubated on grids for 1 hr at room temperature. The EM grids were washed with the incubation solution before treatment with gold conjugate (Protein A-6 nm gold; Aurion). Finally, the grids were washed extensively in PBS (3 × 5 min), 10 nm fiducial gold marker solution was added prior to vitrification in liquid ethane cooled liquid $N_2$.

## Cryo-ET data acquisition and reconstruction

Sixty tilt series of 24 hr grown *C. thermocellum* bacteria cultured on MCC were acquired at 53,000 magnifications, corresponding to a 0.27-nm pixel size, at a defocus of −1 µm. The tilt series cover an angular range of −60° to +60° with an angular increment of 2° and total dose ~150 e⁻/Å² used, using a Thermo Fisher Titan Krios electron microscope operating at 300 keV, equipped with a quantum energy filter, a K2-Summit direct electron detector (Gatan Inc) and a VPP (Thermo Fisher Scientific, Eindhoven, Netherlands).

Tilt series of MCC were acquired at ×53,000 magnification, covering an angular range of −60° to +60° with an angular increment of 2° and total dose ~150 e⁻/Å² and at −6 defocus for visualization purposes.

Interactions of MCC with cellulosomes were analyzed using *C. thermocellum* cultures grown for 15 hr, 23 tilt series at ×53,000 magnification and a defocus of 1–2 µm were acquired using VPP. A single tilt series was recorded using an under-defocus value of 6 µm for visualization purposes (*Figure 2A*). Additional 25 tilt series of 15 hr grown *C. thermocellum* bacteria cultured on cellulose to use for further image processing; hence were acquired at ×105,000 magnification with a corresponding 0.13-nm pixel size, at an under focus of 4 µm, and an angular increment of 3°.

We confirmed the low- and high-density populations were detectable at high-exposure images of ×4800 magnification by comparing it to higher magnification images. The bigger field of view of ×4800 magnification contributed to the visualization of more bacteria and thus, increased the number of bacteria counted. For population analysis, high-exposure images of *C. thermocellum* incubated at five different time periods 15 hr (42 images), 24 hr (219 images), 30 hr (69 images), 48 hr (40 images), and 54 hr (231 images) were acquired as described.

## Image processing
### Population analysis

The total number of high- and low-cellulosome-expressing bacteria (nonlysed and nonsporulating) was manually counted in low-magnification images of *C. thermocellum* for different incubation periods: 84 bacteria for 15 hr, 499 bacteria for 24 hr, 179 bacteria for 30 hr, 170 bacteria for 48 hr, and 230 intact bacteria for 54 hr. The total number of high-cellulosome density-maintaining bacteria was also counted manually for the same images, and percentages of high-cellulosome density-maintaining bacteria were calculated for each condition in an Excel document. The final histogram, providing the ratio of the low- and high-density populations, was plotted in MATLAB (*Figure 3B*).

## Thickness measurements

Central $x$–$y$ slices, ~70 nm in thickness, through 15 bacteria were projected onto a plane. The images were exported to the Fiji software package. The thickness of the given cellulosome layer was measured by subtracting the length from the bacterial S-layer to the proximal cellulosome layer edge from the distal edge of the cellulosome layer. The measured lines were selected to be perpendicular to the S-layer plane. The measurements from each tomogram were analyzed by Excel (*Figure 2—figure supplement 3*). The distance from S-layer to the proximal edge of the cellulosome layer was measured accordingly (*Figure 2—figure supplement 3*).

## Template matching and subtomogram averaging

The in vitro Cel48S structure was low-pass filtered to 50 Å and localized by PyTom template matching (*Hrabe et al., 2012*; *Mahamid et al., 2016*). The globular densities are much smaller and bigger in comparison to Cel48S and the linkers were not assigned with this procedure. The assigned particles were inspected manually in Chimera, and the coordinates that were outside of the cellulosome layer were removed by the Volume Erase tool.

## Mapping the Cel48S structure into tomographic volumes

To visualize enzymatic densities around the bacterium or with the substrate (*Figure 2C,D*), the assigned angles and coordinates obtained from the template-matching procedure were used to demonstrate the in vitro Cel48S structure at the cellulose surface. The mapping of Cel48S units into the tomographic volumes was performed in MATLAB, and figures were prepared in Chimera (UCSF, USA).

## Segmentation

The cellulosomes (*Figure 2B*; *Figure 2—figure supplement 3*, *Video 3*), MCC (*Figure 2B–F* and *Video 3*), enzymes (*Figure 2F*), and bacteria (*Figure 2B*; *Figure 2—figure supplement 3*) were either manually or isosurface-threshold segmented in Amira-Avizo 2019.1 (Thermo Fischer Scientific). All rendered views except *Figure 2—figure supplement 3* were visualized in Chimera (UCSF, USA).

## Confocal microscopy

A volume of 2 ml of cells grown with MCC was centrifuged for 5 min at 5000 × $g$. The cells were washed quickly in TBS and fixed with paraformaldehyde (Sigma-Aldrich, St. Louis, MO) diluted to 4% in TBS for 20 min. The cells were then washed and centrifuged for 5 min at 5000 × $g$, three times with TBS and once in TBS containing 1% Triton X-100 (Sigma-Aldrich). The cells were then resuspended in 5% BSA diluted in TBS for blocking 1 hr at room temperature. After removing the blocking solution by centrifugation, rabbit anti-CBM3a from CipA antibody diluted at 1/5000 in 1% BSA in TBS was added, and the cells were incubated 1 hr at room temperature. The cells were washed three times with TBS by centrifugation as described above, and secondary goat anti-rabbit IgG Alexa 594 antibody (Thermo Fisher Scientific) diluted at 1/1000 in 1% BSA in TBS was allowed to interact for 1 hr at room temperature. After three washes of the cells (by centrifugation as above), the cells were mounted on slides with SlowFade Gold anti-fade reagent (Thermo Fisher Scientific) using polylysine-coated cover slides. Confocal images were obtained by using a LSM880 inverted laser-scanning confocal microscope (Carl Zeiss AG, Oberkochen, Germany) equipped with an Airyscan high-resolution detection unit. A Plain-Apochromat ×63/1.4 Oil DIC M27 objective with a 561-nm DPSS laser with a no-emission filter was used, and parameters were set to avoid pixel intensity saturation and to ensure Nyquist sampling in the $x$–$y$ plane.

## RNA analysis

Three independent sets of cellulose-grown cell cultures were individually sampled at different growth times from 10 to 43 hr. Cells were harvested for 5 min at 3000 × $g$, and the pellets were resuspended in RNA*later* solution according to the manufacturer's protocol. Cells were then stored in −80°C until downstream RNA extraction preparations. The RNA was isolated using an RNAeasy minikit (Qiagen, Germantown, MD) using protocol 4 with the following modifications: cells were incubated 2 hr at room temperature in TBS prepared with RNase-free water supplemented with lysozyme at a concentration of 15 mg/ml and 20 µl proteinase K at 20 mg/ml. Then the purified RNA samples were immediately processed with the Thermo Fisher Scientific verso cDNA synthesis kit. Quantitative real-time PCR

analysis was performed to calculate the RNA levels of two genes of interest: Cel48S and CipA RNA, which were normalized to RecA RNA levels. A specific fragment of each gene was amplified using the primer pairs: 5′-CGCAGAAGGCCGTGCTATA-3′ and 5′-CAGAACCTTTACCCTGCTCCTTT-3′ for Cel48S, 5′-CAGTATGCTCTTAGTTGTGGCTATGC-3′ and 5′-TGATCCAACGGCTGCTGTAA-3′ for CipA, and 5′-GTTGCGGTAAATCTCGATATTGTAAA-3′ and 5′-GGCCAATCTTCTGACCGTTG-3′ for RecA. For each gene, individual standard curves were generated by amplifying serial tenfold dilutions of quantified gel-extracted PCR products obtained by the amplification of each fragment. The standard curves with required standard efficiencies were obtained using six dilution points and were calculated using the Rotorgene 6000 series software (Qiagen). Real-time PCR was performed in a 10-µl reaction mixture containing 5-µl Absolute blue SYBR green master mix (Thermo Fisher Scientific), 0.2 µl of each primer (diluted to 10 µM), 3.6 µl nuclease-free water, and 1 µl of cDNA.

## CipA protein expression analysis

A volume of 2 ml samples from the three independent sets of cellulose-grown cell cultures was taken from times 15, 24, 30, 39, 48, and 54 hr. Half of the volume was used to determine the cell numbers in the culture, by DNA extraction as performed by *Stevenson and Weimer, 2007* without the initial filtration steps, followed by quantitative real-time PCR analysis as described above using 16S universal primers covering the V2 and V3 regions of the gene (5′-TTTGATCNTGGCTCAG-3′ and 5′-GTNTT ACNGCGGCKGCTG-3′).

The calculated number of 16S copies served to normalize the volume taken for western blotting to a similar number of cells. Normalized volumes of samples from the different growth time points were applied on SDS–PAGE gels (10% acrylamide) and transferred to a nitrocellulose membrane using Mini Trans-Blot cells (BioRad Laboratories Ltd, Rishon Le Zion, Israel). Nonspecific protein interactions were blocked by incubating the membrane for 1 hr with 5% BSA diluted in TBST (Tris-buffered saline with 0.05% of Tween 20). Rabbit antibody against CBM3a from CipA (*Morag et al., 1995*) at a dilution of 1:3000 was incubated with the nitrocellulose membrane for 1 hr in 1% BSA diluted in TBST. The membrane was then rinsed three times with TBST and then incubated for 1 hr with secondary anti-rabbit antibody labeled with horseradish peroxidase at a dilution of 1:10,000. The membrane was rinsed again three times with TBST and developed by incubation for 1 min with equal amounts of solutions A and B (ECL, Ornat Biochemicals & Lab Equipment, Rehovot, Israel). Chemiluminescence was quantified using a luminescent image analyzer (Fusion, FX7 G/R/IR).

## Cellulolytic activity per cell

An overnight culture from *C. thermocellum* was used as a starter culture to inoculate 125 ml serum bottles containing 100 ml of GS2media with MCC as the sole carbon source. After inoculation, the serum bottles were incubated at 60°C for 45 hr and samples were taken at 0, 10, 15, 20, 25, 30, 35, 40 and 45 hr. The samples were centrifuged at 4000 rpm for 5 min to separate the bacterial pellet from the supernatant fluids, and the protein fraction was precipitated by adding ammonium sulfate at 80% saturation in order to recover the cellulosomes that were released into the media. Proteinaceous pellets were obtained by centrifuging the solutions from ammonium sulfate treatment at 4000 rpm for 10 min, resuspending the pellets in 1 ml of TBS buffer and then reuniting the resultant solution with the bacterial pellet collected from the original sample (overall ×15-fold concentration). Then, these samples were used for determining the cellulolytic activity per cell using phosphoric acid-swollen cellulose as a substrate (*Wood, 1988*). Enzymatic reactions were performed at 60°C for 72 hr under continuous agitation (600 rpm). After incubation, samples were centrifuged at 14,000 rpm for 10 min, and the supernatant fluids were used to determine the number of released sugars, by the DNS method (*Miller, 2002*). Cell numbers in each sample from different time points were normalized by absolute quantification of the 16S gene copy number using real-time qPCR.

## Effect of soluble sugars on CipA and Cel48S gene expression

In order to determine the effect of soluble sugars on the expression of CipA and Cel48S genes in *C. thermocellum*, either cellobiose or sugars released from MCC by a 20-hr *C. thermocellum* culture were added into the culture media. In the case of cellobiose, a 10 hr growth culture of *C. thermocellum* was supplemented with 10 ml of cellobiose 10% (wt/vol), and samples were taken after 10, 12, 15, and 18 hr. Alternatively, taking into account that the peak of expression of the cellulosomes occurs

at ~15 hr after inoculation, we extracted samples of 0.45-μm filter-sterilized supernatant fluids from a 20-hr growth culture and used it as the culture medium to inoculate the total cells of a 10-hr culture, obtained by centrifuging the cells (and cellulose) for 10 min at 4000 rpm. Before inoculation, cells were washed twice in fresh medium, in order to remove soluble sugar. All the steps were performed under anaerobic conditions. The cultures were incubated at 60°C, and 2 ml samples were taken at 10, 12, 15, and 18 hr. Finally, RNA was analyzed as described above.

## Acknowledgements

This work was funded by the DIP (2476/2-1) to IM and OM, ERC (866530) to IM, ISF (1947/19) to IM and SM, and Swiss National Foundation grant (SNSF 31,003A_179418) to OM. SM is grateful to EMBO for the short-term fellowship (7686) allowing a visit to the University of Zurich. Funding was also provided by the Center for Bioenergy Innovation (CBI), a U.S. Department of Energy Bioenergy Research Center supported by the Office of Biological and Environmental Research in the DOE Office of Science. The authors thank Eva Setter-Lamed (Weizmann Institute of Sciences, Israel) for technical assistance, Matthias Wojtynek (UZH/ETH) for assisting with template-matching procedures, Rebecca de Leeuw (UZH) and Miriam Weber (UZH). The authors thank the center of microscopy and image analysis (ZMB) at the University of Zurich, and the Ilse Katz Institute for Nanoscale Science and Technology Shared Resource Facility under the direction of Dr. Uzi Hadad for image acquisition with a Zeiss LSM880 Airyscan. The authors are also grateful to Prof. Otto X Cordero (MIT) for providing constructive criticisms during the final stages of manuscript preparation.

## Additional information

### Funding

| Funder | Grant reference number | Author |
|---|---|---|
| Deutsche Forschungsgemeinschaft | 2476/2 -1 | Ohad Medalia Itzhak Mizrahi |
| European Research Council | 866530 | Itzhak Mizrahi |
| Israel Science Foundation | 1947/19 | Itzhak Mizrahi Sarah Moraïs |
| Swiss National Science Foundation | 310030_207453 | Ohad Medalia |
| Center for Bioenergy Innovation | | Yannick J Bomble |

The funders had no role in study design, data collection, and interpretation, or the decision to submit the work for publication.

### Author contributions

Meltem Tatli, Sarah Moraïs, Conceptualization, Data curation, Formal analysis, Investigation, Methodology, Validation, Visualization, Writing - original draft; Omar E Tovar-Herrera, Formal analysis, Investigation, Methodology, Writing - review and editing; Yannick J Bomble, Formal analysis, Methodology, Resources, Writing - review and editing; Edward A Bayer, Conceptualization, Validation, Writing - original draft, Writing - review and editing; Ohad Medalia, Conceptualization, Formal analysis, Funding acquisition, Investigation, Methodology, Resources, Supervision, Validation, Visualization, Writing - original draft; Itzhak Mizrahi, Conceptualization, Formal analysis, Funding acquisition, Investigation, Methodology, Resources, Supervision, Validation, Writing - original draft

### Author ORCIDs

Meltem Tatli http://orcid.org/0000-0003-3632-6208
Sarah Moraïs http://orcid.org/0000-0001-9026-2386
Yannick J Bomble http://orcid.org/0000-0001-7624-8000
Ohad Medalia http://orcid.org/0000-0003-0994-2937

Itzhak Mizrahi [iD] http://orcid.org/0000-0001-6636-8818

**Decision letter and Author response**
Decision letter https://doi.org/10.7554/eLife.76523.sa1
Author response https://doi.org/10.7554/eLife.76523.sa2

## Additional files

### Supplementary files
• Transparent reporting form

### Data availability
Structural data that support the findings of this study have been deposited in the Electron Microscopy Data Bank https://www.ebi.ac.uk/emdb/ (accession code EMD-11986). Representative data set can be found in EMPIAR under the accession number EMPIAR-10593.

The following datasets were generated:

| Author(s) | Year | Dataset title | Dataset URL | Database and Identifier |
|---|---|---|---|---|
| Tatli M, Morais S, Tovar-Herrera OE, Bomble YJ, Bayer EA, Medalia O, Mizrahi I | 2021 | Cryo-EM structure of exoglucanase Cel48S | https://www.ebi.ac.uk/emdb/EMD-11986 | Electron Microscopy Data Bank, EMD-11986 |
| Tatli M, Moraïs S, Tovar-Herrera OE, Bomble Y, Bayer EA, Medalia O, Mizrahi I | 2022 | Nanoscale view of the Clostridium thermocellum cellulosome during cellulose degradation reveals an ecological strategy leading to phenotypic heterogeneity | https://www.ebi.ac.uk/empiar/EMPIAR-10593/ | EMPIAR, EMPIAR-10593 |

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
