## [Editor Report]

The premise behind this manuscript is timely and of interest to a broad scientific community working in the field of microbial recycling of cellulosic biomass. It provides a useful link between the occurrence and molecular aspects of the bacterial 'machinery' named cellulosome, and physiological traits of the same bacteria when grown on micro-crystalline cellulose. The key claims of the manuscript are well supported by the data, and the approaches used are thoughtful and rigorous.

---

## [Decision Letter]

**Decision letter after peer review:**

Thank you for submitting your article "Nanoscale resolution of microbial fiber degradation in action" for consideration by *eLife*. Your article has been reviewed by 2 peer reviewers, including Steven Smith as Reviewing Editor and Reviewer #1, and the evaluation has been overseen by Gisela Storz as the Senior Editor. The following individual involved in the review of your submission has agreed to reveal their identity: Mirjam Czjzek (Reviewer #2).

Essential revisions:

1) While the cryo-EM structure of Cel48S allows the template-matching procedure, employed subsequently in the manuscript, it in itself does not represent particularly novel data. As such, this section and associated figures should be shifted to supplemental material. In doing so, the flow and focus of the manuscript will remain entirely on the novel electron tomography data and follow-up studies.

2) The additional comments/suggested edits by the reviewers will also make for a more clear and comprehensive manuscript.

*Reviewer #1 (Recommendations for the authors):*

Considerations for strengthening the manuscript:

1. As written, the section describing the cryo-EM structure of Cel48S, somewhat disrupts the flow of the manuscript by being situated in the "Cellulosome surround the bacteria as an extensive layer of complexes at constant distance from the cell wall" section and "Cellulosome interactions with its substrate". Further, while the cryo-EM structure reveals a 10-amino acid residue section of the linker connecting the catalytic module to the dockerin module that had not previously been observed, this finding and the structure of the catalytic module, which has previously been determined by X-ray crystallography, does not have the novelty/impact of the other data. Consideration could be given to including this data as supplemental with reference at an appropriate place (e.g., line 214 in Cellulosome interactions with its substrate section).

*Reviewer #2 (Recommendations for the authors):*

In my opinion, this work is timely and of great interest to the field of microbial recycling of cellulosic biomass. In my opinion, the only weakness of the manuscript lies in the fact that some of the methods descriptions are particularly short and lack some crucial informative details, making it sometimes complicated to distinguish and understand in-situ experimental set-ups from data obtained from recombinant in-vitro set-ups, and how they correlate; as an example, it does not become clear why a cryo-EM 3D structure of Cel48S was necessary for this study, and why in some (in situ) figures this seems to be the only enzyme present.

Specific comments to the authors:

1. Introduction page2-3: the statement line 78-80, that structural insights into the fiber-degrading process at the nanoscale level remain unknown, is only partially true. A recent study, also entitled the "cellulosome in action", is depicting first insights using AFM. This work by M. Eibinger et al., ACS Cent Sci. 2020 May 27; 6(5): 739-746. doi: 10.1021/acscentsci.0c00050, should be mentioned and cited here somewhere.

2. Page 5 lines 158-159: "We therefore studied the structure of recombinantly expressed full-length Cel48S by cryo-EM."

I imagine that this cryo-EM structure determination was necessary to be able to perform the template-matching procedure, described later in the manuscript. At least in the material and methods description, this should be mentioned (if my assumption is correct). Otherwise, the low(er) resolution cryo-EM structure is not of great interest in the way it is discussed here. From supplemental Figure S3 it becomes clear that only 7% of the single-particle images were used for structural refinement; it would be interesting to know if the other particles were omitted only because of lack of resolution/quality or if the low percentage of identical images is due to inherent flexibility of the enzymes, or part of the enzyme (i.e specific loops or sub-domains?)? Are the other images just different and lower resolution "views/orientations", or do they contain any additional information that could be of interest (as compared to a more rigid crystal structure, at a higher resolution but tightly packed within the crystal lattice)?

3. The title of the Figure legend for Figure 2 is misleading: "Structure of Cel48S and location of cellulosomal enzymes around the C. thermocellum cell".

While I do see the cryo-EM structure of Cel48S depicted in panels A to D, I do not see any illustration of the location of the enzymes around the C. thermocellum cell in this figure!

4. Page 7 line 206: "The cellulose was identified within the dense cellulosome layer,…"

How were the cellulose fibers identified? Is it 'only' visual inspection? Do the cryo-ET images allow determining density values that can be correlated to different types of molecules/objects/molecular structures?

5. Page 7 line 211: "…, we employed a template-matching procedure, as conducted above."

This must be a typo; it is in fact the first time template matching is mentioned in this manuscript. Please correct and replace "as conducted above" with the exact location in the manuscript where this is precisely described.

6. Figure 3 C and D. It is a bit intriguing that the selected portion showing presumably Cel48S enzymes in contact with cellulose micro-fibrils of the in situ study matches extremely well with the more directed in vitro image of Cel48S (only) containing mini-cellulosomes in interaction with microfibrils. Indeed, one would expect to have globular enzymes of variable size in the in-situ image, since the "natural cellulosome" does contain other enzymes than Cel48S, although proven to be a major component. Also, where is the rest of the cellulosomal proteins located in this view? Why nothing resembling a scaffoldin can be seen in close vicinity? Are the parts to that extent flexible (through the linkers) that the rest of the machinery comes to lie so distantly that they are not caught in this close-up view?

7. In the same idea, it would be interesting to describe, and for the reader to know, if the template matching of Cel48S works in all places? Or do globular regions that cannot be matched with Cel48S occur? Page 17, in the methods description line 596, the authors state that 63 Cel48S enzymes can be assigned in a sub-volume shown in Figure 3C. Thus, are there regions (and how many) that can not be assigned to Cel48S?

Also the sentence: "The final number of assigned 63 Cel48S for the subvolume (Figure 3C)" needs to be reformulated or corrected; as it stands it is not clear.

8. Figure 3F. From the text, it is not clear what "treatment" has been done to the data to "extract" the altered microcellulose fibrils, shown in these rendered illustrations. In the raw data, are cellulosomal particles present and identified in the regions of the kinks that are highlighted by the arrowheads? Why not present the figure with a cellulosomal particle in orange, at least for one of these extractions (as it appears in one of the videos)? It would render the figure much clearer.

---

## [Author Response]

Essential revisions:1) While the cryo-EM structure of Cel48S allows the template-matching procedure, employed subsequently in the manuscript, it in itself does not represent particularly novel data. As such, this section and associated figures should be shifted to supplemental material. In doing so, the flow and focus of the manuscript will remain entirely on the novel electron tomography data and follow-up studies.

We agree with these suggestions and shifted this section and the respective figure of the single particle cryo-EM in Figure 2 – it has now been transferred to Figure 2—figure supplement 2.

2) The additional comments/suggested edits by the reviewers will also make for a more clear and comprehensive manuscript.

We thank again the reviewers and followed their suggestion.

Reviewer #1 (Recommendations for the authors):Considerations for strengthening the manuscript:1. As written, the section describing the cryo-EM structure of Cel48S, somewhat disrupts the flow of the manuscript by being situated in the "Cellulosome surround the bacteria as an extensive layer of complexes at constant distance from the cell wall" section and "Cellulosome interactions with its substrate". Further, while the cryo-EM structure reveals a 10-amino acid residue section of the linker connecting the catalytic module to the dockerin module that had not previously been observed, this finding and the structure of the catalytic module, which has previously been determined by X-ray crystallography, does not have the novelty/impact of the other data. Consideration could be given to including this data as supplemental with reference at an appropriate place (e.g., line 214 in Cellulosome interactions with its substrate section).

As mentioned above, the figure was shifted into the supplementary figures, and the paragraph was removed from the main text.

Reviewer #2 (Recommendations for the authors):In my opinion, this work is timely and of great interest to the field of microbial recycling of cellulosic biomass. In my opinion, the only weakness of the manuscript lies in the fact that some of the methods descriptions are particularly short and lack some crucial informative details, making it sometimes complicated to distinguish and understand in-situ experimental set-ups from data obtained from recombinant in-vitro set-ups, and how they correlate; as an example, it does not become clear why a cryo-EM 3D structure of Cel48S was necessary for this study, and why in some (in situ) figures this seems to be the only enzyme present.Specific comments to the authors:1. Introduction page2-3: the statement line 78-80, that structural insights into the fiber-degrading process at the nanoscale level remain unknown, is only partially true. A recent study, also entitled the "cellulosome in action", is depicting first insights using AFM. This work by M. Eibinger et al., ACS Cent Sci. 2020 May 27; 6(5): 739-746. doi: 10.1021/acscentsci.0c00050, should be mentioned and cited here somewhere.

We thank the reviewer for this comment. Here, cryo-ET technology allowed the 3D reconstruction of the cellulosome layer at ~2 nm resolution, which is a different scale than that described in the AFM work. We thus modified our sentence in the introduction and refer to this work in the discussion.

2. Page 5 lines 158-159: "We therefore studied the structure of recombinantly expressed full-length Cel48S by cryo-EM."I imagine that this cryo-EM structure determination was necessary to be able to perform the template-matching procedure, described later in the manuscript. At least in the material and methods description, this should be mentioned (if my assumption is correct). Otherwise, the low(er) resolution cryo-EM structure is not of great interest in the way it is discussed here. From supplemental Figure S3 it becomes clear that only 7% of the single-particle images were used for structural refinement; it would be interesting to know if the other particles were omitted only because of lack of resolution/quality or if the low percentage of identical images is due to inherent flexibility of the enzymes, or part of the enzyme (i.e specific loops or sub-domains?)? Are the other images just different and lower resolution "views/orientations", or do they contain any additional information that could be of interest (as compared to a more rigid crystal structure, at a higher resolution but tightly packed within the crystal lattice)?

As suggested above for the comments of Reviewer 1, the description of the single particle cryo-EM was transferred to the supplementary section together with the relevant paragraph.

In cryo-EM, a large number of particle images are acquired, while many do not contain high resolution information. In some cases, the structures are determined from an even smaller relative population e.g., in Sukalskaia et al., 2021 (Sukalskaia, A., Straub, M.S., Deneka, D. et al. Cryo-EM structures of the TTYH family reveal a novel architecture for lipid interactions. Nat Commun 12, 4893 (2021). https://doi.org/10.1038/s41467-021-25106-4) out of 3.5M particles, 267K were used in the final reconstruction, 7%.

The particles that were removed on the way were clearly due to low resolution. This may be due to flexibility as well as due to local environment within the ice. In the 2D classes we could not identify any extra regions that could have been attributed to the additional subdomain or specific loops. Therefore, we cannot comment further regarding the reason for which many particles were omitted during the procedure.

3. The title of the Figure legend for Figure 2 is misleading: "Structure of Cel48S and location of cellulosomal enzymes around the C. thermocellum cell".While I do see the cryo-EM structure of Cel48S depicted in panels A to D, I do not see any illustration of the location of the enzymes around the C. thermocellum cell in this figure!

We thank the reviewer for pointing out this error, that was now corrected and shifted to the supplementary material.

4. Page 7 line 206: "The cellulose was identified within the dense cellulosome layer,…"How were the cellulose fibers identified? Is it 'only' visual inspection? Do the cryo-ET images allow determining density values that can be correlated to different types of molecules/objects/molecular structures?

The microcrystalline cellulose has a clear identifiable appearance in cryo-ET. We studied pure MCC as an initial step, and we thus learned its characteristic density and appearance. The density differences and appearance between the MCC and the globular enzymatic densities and the flexible scaffoldins between the globular enzymes are very clear. The stacking of the polysaccharide fibers (MCC) is clearly different that the protein appearance and electron scattering.

5. Page 7 line 211: "…, we employed a template-matching procedure, as conducted above."This must be a typo; it is in fact the first time template matching is mentioned in this manuscript. Please correct and replace "as conducted above" with the exact location in the manuscript where this is precisely described.

We thank the reviewer for finding this error. We corrected this mistake.

6. Figure 3 C and D. It is a bit intriguing that the selected portion showing presumably Cel48S enzymes in contact with cellulose micro-fibrils of the in situ study matches extremely well with the more directed in vitro image of Cel48S (only) containing mini-cellulosomes in interaction with microfibrils. Indeed, one would expect to have globular enzymes of variable size in the in-situ image, since the "natural cellulosome" does contain other enzymes than Cel48S, although proven to be a major component. Also, where is the rest of the cellulosomal proteins located in this view? Why nothing resembling a scaffoldin can be seen in close vicinity? Are the parts to that extent flexible (through the linkers) that the rest of the machinery comes to lie so distantly that they are not caught in this close-up view?

The reviewer is correct. At the resolution of the intact bacterial tomogram, it is challenging to distinguish between the Cel48S and other similar size cellulosome enzymes. Therefore, it cannot be excluded that the template matching procedure also identified some of the other globular cellulosome enzymes, around the MCC. However, the most abundant enzyme in the cellulosome is the Cel48S. Additionally, the binding of pure Cel48S to MCC mimics well the in situ binding properties of the cellulosome enzymes, presumably due to the high binding affinity of the enzymes to their substrate. It is also important to mention that proteins that are substantially smaller or larger than Cel48S, as well as of different shapes, e.g. elongated structure of scaffoldins, were not matched in this procedure. For example, the physical characteristics of the globular cellulolytic enzymes are of relatively high contrast compared to the flexible rod-like appearance of scaffoldins. This is the primary reason that we detected the globular density of the enzymes rather than the scaffoldins.

In the rendering views we labeled manually the MCC while the globular densities were detected automatically, however, also manual segmentations are shown (Figure 2—figure supplement 4, B and C). In some cases, elongated protein densities are seen between globular enzymes within the cellulosome layer which may resemble scaffoldins, however, they are challenging to track due to their flexibility and limited contrast. In addition, no high resolution structures of these proteins are available for comparison using template matching.

To summarize, we are confident that the globular structures detected correspond to the cellulosome enzymes and that Cel48S is the most abundant enzyme of the cellulosomes (Raman et al., 2011; Morag et al., 1991; Zverlov et al., 2005; Yoav et al., 2017). We also find the high resemblance between the in vitro and in situ structures overwhelming. However, in this resolution of the individual tomogram, we cannot distinguish the Cel48S from the other large globular enzymes of the cellulosome. Therefore it is likely that most of the density we found are of Cel48S enzyme while residual other cellulosomal enzymes may have detected as well. This was clarified in the text.

7. In the same idea, it would be interesting to describe, and for the reader to know, if the template matching of Cel48S works in all places? Or do globular regions that cannot be matched with Cel48S occur? Page 17, in the methods description line 596, the authors state that 63 Cel48S enzymes can be assigned in a sub-volume shown in Figure 3C. Thus, are there regions (and how many) that can not be assigned to Cel48S?

Thank you for this comment and questions, we hope that we answered the question in the above paragraph, we added the above information to the text discussion the better clarify these points.

Also the sentence: "The final number of assigned 63 Cel48S for the subvolume (Figure 3C)" needs to be reformulated or corrected; as it stands it is not clear.

The text has been removed.

8. Figure 3F. From the text, it is not clear what "treatment" has been done to the data to "extract" the altered microcellulose fibrils, shown in these rendered illustrations. In the raw data, are cellulosomal particles present and identified in the regions of the kinks that are highlighted by the arrowheads? Why not present the figure with a cellulosomal particle in orange, at least for one of these extractions (as it appears in one of the videos)? It would render the figure much clearer.

The MCC was simply rendered from the tomograms. We mention this the Materials and methods section. MCC structures, decorated with cellulosomes, are shown in Figure 2F- Figure 2—figure supplement 4C-D.